# Developing Iron Nanochelating Agents: Preliminary Investigation of Effectiveness and Safety for Central Nervous System Applications

**DOI:** 10.3390/ijms25020729

**Published:** 2024-01-05

**Authors:** Eleonora Ficiarà, Chiara Molinar, Silvia Gazzin, Sri Jayanti, Monica Argenziano, Lucia Nasi, Francesca Casoli, Franca Albertini, Shoeb Anwar Ansari, Andrea Marcantoni, Giulia Tomagra, Valentina Carabelli, Caterina Guiot, Federico D’Agata, Roberta Cavalli

**Affiliations:** 1School of Pharmacy, Center for Neuroscience, University of Camerino, 62032 Camerino, Italy; eleonora.ficiara@unicam.it; 2Department of Drug Science and Technology, University of Turin, 10125 Turin, Italy; chiara.molinar@unito.it (C.M.); monica.argenziano@unito.it (M.A.); andrea.marcantoni@unito.it (A.M.); giulia.tomagra@unito.it (G.T.); valentina.carabelli@unito.it (V.C.); roberta.cavalli@unito.it (R.C.); 3Fondazione Italiana Fegato-Onlus, Bldg. Q, AREA Science Park, ss14, Km 163.5, Basovizza, 34149 Trieste, Italy; silvia.gazzin@fegato.it (S.G.); sri.jayanti@fegato.it (S.J.); 4Institute of Materials for Electronics and Magnetism (IMEM) CNR, Parco Area delle Scienze 37/A, 43124 Parma, Italy; lucia.nasi@imem.cnr.it (L.N.); francesca.casoli@imem.cnr.it (F.C.); franca.albertini@imem.cnr.it (F.A.); 5Department of Neurosciences, University of Turin, 10124 Turin, Italy; shoeb.ansari@edu.unito.it (S.A.A.); caterina.guiot@unito.it (C.G.)

**Keywords:** nanobubbles, iron, chelation, neurodegeneration, DFO

## Abstract

Excessive iron levels are believed to contribute to the development of neurodegenerative disorders by promoting oxidative stress and harmful protein clustering. Novel chelation treatments that can effectively remove excess iron while minimizing negative effects on the nervous system are being explored. This study focuses on the creation and evaluation of innovative nanobubble (NB) formulations, shelled with various polymers such as glycol-chitosan (GC) and glycol-chitosan conjugated with deferoxamine (DFO), to enhance their ability to bind iron. Various methods were used to evaluate their physical and chemical properties, chelation capacity in diverse iron solutions and impact on reactive oxygen species (ROS). Notably, the GC-DFO NBs demonstrated the ability to decrease amyloid-β protein misfolding caused by iron. To assess potential toxicity, in vitro cytotoxicity testing was conducted using organotypic brain cultures from the substantia nigra, revealing no adverse effects at appropriate concentrations. Additionally, the impact of NBs on spontaneous electrical signaling in hippocampal neurons was examined. Our findings suggest a novel nanochelation approach utilizing DFO-conjugated NBs for the removal of excess iron in cerebral regions, potentially preventing neurotoxic effects.

## 1. Introduction

Excess iron in aging is considered a possible trigger of neurodegenerative diseases [1]. There is evidence for iron accumulation in the brains of older individuals and Alzheimer’s disease (AD) patients, and alterations in iron levels in different biofluids, such as cerebrospinal fluid (CSF) [2]. 

Iron is involved in AD genesis and progression under several different pathways. Iron trivalent ions play a key role in AD development by binding to Amyloid β protein (Aβ), favoring its misfolding and aggregation [3].

Indeed, fibrillogenesis requires the partial unfolding of globular proteins. Additionally, other amyloidogenic proteins such as α-synuclein and tau are prone to fibrillation, despite being natively unfolded. Iron can also enhance the aggregation process of α-synuclein which is induced in the development of Parkinson’s disease (PD) [4]. How iron enhances protein aggregation is not fully understood, but two distinct mechanisms are considered relevant: first, the direct binding of iron to amyloidogenic proteins, and second, an indirect iron-mediated process, where the Fenton and Haber–Weiss reactions of Fe^2+^ trigger aggregation through reactive oxygen species (ROS) production and the resulting oxidative stress [5]. Moreover, Fe^3+^ is also responsible for oxidative damage due to the Fenton and Haber–Weiss reactions, producing highly reactive ROS [6].

Due to its high redox potential, iron is involved in oxidative stress related to AD. A previous study demonstrated that redox-active iron in CSF positively correlates with the degree of cognitive impairment in normal to mildly cognitively impaired subjects, whereas redox-active iron in CSF is dramatically reduced to a level close to zero in AD [7].

In addition, iron overload has been implicated in the pathology and pathogenesis of PD [8]. The substantia nigra, where the selective loss of dopaminergic neurons occurs, is the primary region in the brain where iron deposits have been found, together with other cerebral regions involved, e.g., the basal nuclei [9,10].

Therefore, in this context, metal chelation therapies in neurodegenerative diseases have attracted growing interest [11].

Iron chelation has been introduced as a new therapeutic strategy for the treatment of neurodegenerative diseases that have a component of metal ion accumulation to remove excess toxic iron in patients [12].

Interestingly, iron chelators have attracted much research interest due to their possible therapeutic value in eliminating excess iron from certain brain regions and showing their ability to prevent or even reverse Aβ protein aggregation [13]. Clinical demonstrations of chelation removing labile iron accumulated in specific brain areas implicated in neurodegenerative disease are also available [14].

Deferoxamine (DFO) is the first iron chelator drug approved by the FDA to treat excess iron and is the iron chelator with the longest clinical application in treating iron overload. DFO can form very stable complexes, wrapping around iron at six coordination sites [15]. Moreover, it is a highly specific chelator for ferric iron and has very little affinity for ferrous iron. DFO has been used for iron chelation therapy in transfusion-dependent thalassemia patients [16]. However, its inherent pharmacokinetic and pharmacodynamic shortcomings, such as a short plasma half-life and cytotoxicity, need to be addressed to widen its clinical utility [17].

To overcome its limitations, DFO has been conjugated to dendrimers, polymers and nanostructures to obtain DFO-conjugated nanoparticles or nanochelators [18,19].

A water-soluble chitosan-based polymer exhibiting both antioxidant and chelating properties due to dual functionalization with DOTAGA and DFO has been developed to extract this iron, therefore preventing its catalytic production of reactive oxygen species [20].

Engineered DFO nanoparticles were shown to be potentially effective for novel chelator therapy and to effectively target the central nervous system (CNS). An ideal chelating agent entering the CNS should be preferably of ‘nano’ dimensions to facilitate the penetration of brain barriers (provided proper functionalization or physical forcing mechanisms are applied), safe and not toxic, locally deliverable (e.g., intrathecally via CSF), monitorable with available clinical equipment, and potentially drivable towards the precise target.

In our group, we focused on nanobubbles (NBs), small vesicle-like structures consisting of a gas or vaporizable compound core stabilized by a polymeric shell, loadable with drugs and purposely functionalized (e.g., a chelating agent). Besides their versatility, NBs can be visualized by clinical ultrasound (US) imaging (sonography), due to their inner core consisting of a gas or a vaporizable compound. Therefore, they are possible theranostic tools suitable for different applications. NBs have attracted much research attention because they can overcome the limitations of microbubbles currently used in clinical practice as ultrasound contrast agents [21]. The synthesis of DFO functional and biocompatible polymers for use as bioconjugates in the field of polymer therapeutics has received increasing attention in recent years [22].

Based on these premises, the aim of the present work is the development of iron-chelating NBs able to increase DFO’s effectiveness and half-life, decreasing metabolic degradation and toxicity issues. Here, DFO was conjugated to glycol-chitosan (GC). The GC-DFO conjugate was characterized and used to formulate new iron-chelating NBs. GC was selected for its stability, biocompatibility and chelating potential due to the presence of o hydroxyl groups distributed within the polymer chain [23]. 

We therefore manufactured two nanoformulations, using GC and GC-DFO to optimize their chelation capacity.

Finally, the safety in the CNS using such formulations of NBs was investigated in vitro on rodent cultured cells from the substantia nigra and hippocampus, showing that, at proper concentrations, the neural network was neither seriously damaged in its structure nor in its bioelectric functionality.

## 2. Results

DFO has a linear structure (Figure 1a) and can wrap around iron, chelating it at all six coordination sites (Figure 1b).

The structure of the GC-DFO conjugate was assessed both through FT-IR (Figure 2a) and NMR spectroscopy (Figure 2b).

The disappearance and the shifting of peaks in the spectroscopic profile of GC and DFO demonstrate the modification of the polymer chain. The same behavior was observed in the NMR spectra.

### 2.1. Physicochemical Characterization of the Investigated Nanoformulations

Table 1 summarizes the physicochemical characterization of all the NBs formulations. All the NBs showed an average diameter lower than 300 nm and a negative surface charge when shelled with GC conjugated with DFO.

The viscosity (mean ± SD) was 1.35 ± 0.02 cP for GC-NBs and 1.30 ± 0.01 cP for the GC-DFO NB aqueous nanosuspension, respectively. 

TEM confirmed the spherical morphology and the core–shell structure of the GC-DFO NB (Figure 3). The original NB dimension was slightly modified with a small diameter increase, due to the interaction of the TEM-related electronic energy causing inner gas vaporization with an enhancement in NB volume.

NBs in both formulations, with and without DFO, can be observed in the sonography results, showing echogenic properties (Figure 3).

Following their administration the chelation capability of the iron content of the investigated formulations (DFO, GC-DFO conjugate, GC NBs and GC-DFO NBs) was assessed, confirming their ability to chelate Fe^3+^ to a different extent and form a chelation complex (Figure 4). The GC-DFO conjugate maintains the marked chelation ability of free DFO. Interestingly, the organization of the polymeric GC-DFO shell on the NB surface confers to the polymer chains a conformation suitable to interact with Fe^3+^, enhancing their iron chelation capacity. The remarkable chelation capacity of the GC-DFO NB could also be due, especially at low iron concentrations, to the chelating properties exhibited by the GC NBs. Significant statistical differences between the different formulations at each iron concentration (*p* < 0.0001, two-way ANOVA followed by Tukey’s multiple comparisons test) were detected, except for GC-DFO vs. DFO at 25 µM, 100 µM and 800 µM and for GC-DFO vs. NB GC-DFO at 1600 µM and 3200 µM.

Besides their ability to chelate excess iron, it is of paramount importance to consider the impact of chelating NBs on the secondary effects of iron in pathologies.

Since it is well known that iron promotes the misfolding of amyloid proteins, which progresses towards the formation of toxic oligomers and fibrils, the possible role played by chelating NBs should be investigated. Thioflavin T was used to detect the presence of amyloid aggregates (Aβ_1–42_). A high-thioflavin T fluorescence signal was observed in the presence of iron chelators (DFO, GC-DFO and NB GC-DFO) concerning Fe^3+^, confirming that iron chelation plays a crucial role in preventing Aβ misfolding (Figure 5).

### 2.2. ROS Formation

As expected, considering that Fe^3+^ is involved in the generation of ROS, upon adding Fe^3+^ to the solution, the measured ROS concentration was 1.10 ± 0.30 µM. When both NB formulations (GC-DFO) and Fe^3+^ were added (see Section 4.2.8), the ROS concentration in the sample dropped to 0.01 ± 0.01 µM. 

### 2.3. Assessment of the Safety of the Nanoformulations

#### Hemolytic Activity

The GC-DFO NB aqueous nanosuspensions examined at various concentrations exhibited no significant hemolytic activity lower than 4%, resulting in negligible red blood cell destruction as compared to the control (100% hemolysis). These findings indicate that these nanoformulations are biocompatible and acceptable for prospective in vivo delivery without hemolysis.

### 2.4. Evaluation of NB Safety in Organotypic Brain Culture–Substantia Nigra Slices

When slices of organotypic brain culture–substantia nigra (OBCs-SN) were exposed to GC-DFO NB formulations (biological repetitions: N = 6), no increases in MTT test vs. DMSO exposed slices (positive control) were observed, indicating that all of them were not toxic for mitochondria. Different dilutions corresponded to different concentrations of DFO: 1:8 = 125 μg/mL, 1:64 = 15.62 μg/mL and 1:192 = 5.21 μg/mL. After Fe^3+^ chelation, GC-DFO NBs showed a significant reduction (30%, *p* ≤ 0.01) in the cellular response to 1:8 dilutions (i.e., possible mitochondrial damage). On the contrary, iron-chelated GC-DFO at 1:64 (*p* ≤ 0.001) and 1:192 dilutions (*p* ≤ 0.01) significantly increased the MTT signal, and no mitochondrial damage was detected (see Figure 6a).

The LDH test, indicative of membrane leakage, revealed significant toxicity of GC-DFO at the minimal dilution (1:8, both *p* ≤ 0.05). The toxicity of iron-chelated GC-DFO preparations at 1:8 dilution (*p* ≤ 0.05) was also confirmed by the LDH test (see Figure 6b). 

Because DOPA neurons, responsible for motor impairment in PD, are a neuron family especially vulnerable to stressors, we finally evaluated the potential toxicity of the NB formulations by counting the percentage of surviving DOPA neurons after their administration. As depicted by Figure 6c, GC-DFO NBs significantly reduced DOPAns at both 1:8 and 1:64 dilutions (both *p* ≤ 0.05). At all dilutions, the iron-chelated GC-DFOs were toxic to DOPA, decreasing their number to a level comparable with those observed in slices treated with rotenone (Rot), which mimics the DOPA loss in PD (1:64 and 1:192, both *p* ≤ 0.05 vs. DMSO) or even surpasses it (1:8, *p* ≤ 0.001 vs. DMSO). A dose-dependent trend was also observable in GC-DFO post-chelation, with 1:8 dilution being significantly toxic to DOPAns (*p* ≤ 0.05, see Figure 6c). No significant differences were found between DMSO and untreated slices, indicating the suitability of DMSO as a positive control.

### 2.5. Effects of GC-DFO NBs and Iron on Spontaneous Firing of Hippocampal Network

Given the effectiveness of NBs in counteracting the oxidant action of Fe and the effect of Fe on Aβ misfolding (Figure 5), we next investigated whether Fe alone or in combination with NBs could be responsible for functional alterations in a primary cultured hippocampal network. In this regard, we monitored the spontaneous electrical activity of hippocampal neurons using MEAs (Figure 7a,b) observing that, as previously shown [24], after 18 DIVs, the mature network generated synchronous bursts of action potentials. We first characterized whether NBs (1:190 dilution) affected spontaneous firing and burst activity, observing no significant differences between the control and treated neurons. In this regard, the mean firing frequency in the control condition was 1.44 ± 0.16 Hz (n = 134 electrodes, Nmea = 3) vs. 1.59 ± 1.18 Hz (n = 133 electrodes, Nmea = 3, *p* > 0.05) in the presence of NBs (not shown), while the number of bursts generated in the control neurons was 11.72 ± 0.54 (n = 138 electrodes, Nmea = 3) and in the presence of NBs was 13.25 ± 0.99, (n = 137 electrodes, Nmea = 3, *p* > 0.05).

After proving that NBs alone do not interfere with spontaneous neuronal firing, we next focused on the effect of iron (30 μM), observing that after 5 h of treatment, the mean firing frequency was comparable to that observed in the control conditions (2.95 ± 0.33 Hz in ctrl vs. 2.99 ± 0.41 Hz with Fe, n = 199 electrodes, Nmea = 7, *p* > 0.05, Figure 7a–c), while after 24 h from Fe administration, the firing frequency decreased significantly to 0.81 ± 0.13 Hz (*p* < 0.05). Interestingly, when Fe was administered together with NBs, we observed that after 24 h of treatment, the average firing frequency recovered significantly to 2.03 ± 0.48 Hz (n = 78 electrodes, Nmea = 3, *p* < 0.05). Comparable results were obtained when the number of bursts was considered. In this regard, we observed that after 5 h from Fe administration, the number of bursts measured during a 120 s recording decreased significantly from 19.19 ± 1.77 to 15.52 ± 2.64 (*p* < 0.05) and that this value was further reduced to 3.30 ± 1.13 bursts after 24 h (*p* < 0.05, Figure 7d). Interestingly, when neurons were treated with Fe together with NBs for 24 h, the number of bursts recovered significantly to 9.93 ± 1.87 bursts (*p* < 0.05). These results suggest that NBs are a valuable tool for preserving neuronal function following impairments induced by Fe administration.

## 3. Discussion

In this work, we proposed an advanced nanobubble formulation with iron chelating capacity, investigating the main physicochemical properties and the relative safety for potential applications in neurodegenerative diseases. 

GC was selected as the NB polymer shell because it is soluble in water and not cytotoxic.

For the manufacturing of the chelating NBs, we used conjugated GC and DFO, with DFO representing a chelating agent that is widely studied due to its suitability for several clinical and analytical applications in Fe^3+^ chelation therapy and its versatility in the most recent applications in sensing [25]. 

All these studies are preliminary to the investigation of how the proposed nanostructure could be delivered and functionalized to penetrate the brain barriers, i.e., the BBB in the case of systemic administration or the choroid plexus membranes in the case of intrathecal administration. Stable GC-DFO-shelled NBs were obtained with increased effectiveness in iron chelation compared to that of free DFO. 

The shift from positive to negative values of the zeta potential represented the GC and GC-DFO fingerprints of the polymer on the NB surface, respectively (see Table 1). Indeed, the change in the surface charge value demonstrated the presence of DFO on the outer NB layer. After iron chelation, the NB surface charge remarkably decreased (by about 40%), showing a decrease in DFO units available. Interestingly, the size of the GC NBs decreased after iron chelation (see Table 1). This decrease could indicate the packing of the GC-DFO polymer chains due to the electrostatic interaction with the iron ions. This behavior has already been observed with other polysaccharide shells [26].

The morphological structure of GC-DFO NBs can be clearly visualized by TEM, showing the core–shell structure (see Figure 3). Also, their echogenic properties can be assessed by sonography (see Figure 3), demonstrating the visualization capability of the nanoplatform, which can be exploited for clinical application.

The chelation capacity was assessed through colorimetric assays, showing that the GC-DFO NBs were able to chelate iron, even more than DFO alone, especially at low iron concentrations (see Figure 4). In fact, due to a surface organization that favors electrostatic interaction between the iron and hydroxyls of DFO, a stable complex is formed which enhances the synergy of GC and DFO. The absence of hemolytic activity in NBs is a key parameter for the safety of a nanocarrier and its biocompatibility and is strictly required for intravenous administration in early preclinical development.

The recent development of a deferoxamine-based nanochelator with selective renal excretion has shown promise in ameliorating animal models of iron overload with a substantially improved safety profile [27].

Silica nanoparticles with chelating properties have already been proposed to remove iron from biological media [13], but as far as we know, no specific applications to the brain and the CNS have been presented so far. Although chelating free iron in the brain is a tempting strategy, limitations must be considered. Iron is an essential cofactor in multi-fold cellular processes. Therefore, iron chelation can have off-target effects and potentially cause untoward effects [28,29].

The antioxidant activity of the different formulations of NBs was evaluated, showing that NB chelating iron reduced the percentage of ROS with respect to the free iron solution. This result could indicate the potential use of NBs in regulating the redox state, reducing the damage caused by iron excess, typical of several neurodegenerative diseases. Several works in the literature lay the foundation for antioxidant-based nanotherapeutic candidates attenuating oxidative stress, with potential applications in treating and preventing neurodegenerative conditions [30,31].

In addition, NBs’ ability to counteract iron-induced Aβ protein misfolding was tested with a thioflavin assay, showing that their chelating activity impacted one of the main pathological mechanisms responsible for AD progression. 

Moreover, MTT, LDH and an immunofluorescence cytotoxicity assay were performed on an organotypic brain cell culture to assess the safety of NBs. We confirmed that our NB formulation of GC-DFO (pre/post-chelation) at the higher dilution tested (1:192 *v*/*v*, corresponding to 5.2 mg/mL of DFO) is safe.

This may suggest that at 1:8 *v*/*v* dilution, corresponding to 125 mg/mL of DFO, the number of NBs that can enter the cells is very high, thus damaging their mitochondrial functions or even “engulfing” the cells. 

Interestingly, to investigate in detail iron’s contribution to many neuropathologies, well-designed model systems (in vitro and ex vivo) are required to detect iron movement and ongoing damage at a cellular level [32]. Several studies were conducted with cell lines of neurons, microglia and oligodendrocytes, highlighting that single-cell types are not representative of the natural neuronal network. Organotypic cultures are efficient and reliable ex vivo models which preserve the complex brain milieu present in in vivo studies, combining the accessibility and convenience of in vitro models [33].

Therefore, a strength of our experiments is that OBCs can better replicate the intricate cellular relationships of the brain with respect to monocultures. It is noteworthy that OBCs have served as valuable ex vivo models for exploring the therapeutic potential of specific compounds in neurological diseases like brain tumors, AD and PD [34,35]. Research on GC-based nanoparticles in ex vivo models has been carried out as well using primary olfactory ensheathing cells [36]. Despite this, to the best of our knowledge, there is a notable absence of studies investigating GC-DFO nanoparticles in OBCs, representing an intriguing avenue for our investigation.

Another possible toxicity mechanism considered in this study is the alteration of the spontaneous electrical signaling among neurons that normally occurs in the healthy brain. Accurate electrophysiological measurements on hippocampal neurons showed that iron severely affected their spontaneous firing and burst activity, which, however, was partially recovered following the administration of GC-DFO NBs. 

## 4. Materials and Methods

Ethanol, perfluoropentane, ferrous chloride tetrahydrate (FeCl_2_·4H_2_O) and ferric chloride hexahydrate (FeCl_3_·6H_2_O) were obtained from Sigma-Aldrich (St. Louis, MO, USA). Epikuron^®^ 200 (soy phosphatidylcholine 95%) was a kind gift from Cargill (Wayzata, MN, USA). Palmitic acid was obtained from Fluka (Buchs, CH, Switzerland).

### 4.1. Synthesis of GC-DFO

GC was dissolved in 10 mL PBS (pH 7.4). A total of 1 mL of a solution of succinic anhydride (250 mg/mL) pre-dissolved in DMSO was added to the solution of GC with stirring at rt for 30 min, producing GC-COOH. A negative result of the ninhydrin test confirmed the conversion of the amines to carboxylates.

A DFO solution (80 µmol; 1 eq., 52 mg) was added to the resulting GC-COOH (165 µmol; 2 eq.) in the presence of 4-(4,6-Dimethoxy-1,3,5-triazin-2-yl)-4-methylmorpholinium chloride (DMTMM) (88 µmol; 1.1 eq) under vigorous stirring at 60 °C for 1 h.

The obtained GC-DFO was centrifugated, resuspended and washed several times with the addition of acetone/EA (4/1, *v*/*v*%, >100 mL). The sample was dried and subsequently lyophilized.

### 4.2. Structural Characterization of Synthesized GC-DFO Conjugate 

#### 4.2.1. IR and NMR Spectroscopy

The structure of GC-DFO was determined using the FT-IR Technique. This was carried out by molecules absorbing infra-red light in the region of 4000–650 cm^−1^. The change in energy was measured by observing the molecules’ vibration. Potassium bromide (KBr) discs with all the samples were prepared using an electrically operated KBr press model. The FT-IR spectra of GC, DFO and GC DFO, were recorded on a PerkinElmer 100 FTIR spectrometer using an attenuated total reflectance (ATR) accessory. All the samples were scanned from 4000 to 650 cm^−1^ at a resolution of 4 cm^−1^ and 8 scans/spectrum.

NMR spectra were acquired with a Bruker Avance-600 Ultrashield spectrometer equipped with a 5 mm TBI S3 probe with a Z gradient and variable temperature capability. Samples were prepared at 5 mg/mL.

#### 4.2.2. Preparation of NB Formulations

Epikuron 200^®^ (1.0% *w*/*v*) and palmitic acid (1.0% *w*/*v*) ethanolic solutions were prepared. In an ice bath, 300 µL of Epikuron and palmitic acid solution was added to 450 µL of decafluoropentane. Then, drop by drop, the appropriate amount of water was added to the mixture. A high-shear homogenizer (Ultra-Turrax, IKA-Werke, Staufen, Germany) was used to homogenize the system for 2 min at 16,000 rpm until a nanoemulsion formed. NB formulations were obtained through the dropwise addition of 300 µL of 2.0% GC or 2.0% GC–DFO. After the preparation, NB formulations were purified through the dialysis technique using a cellulose dialysis membrane with a cut-off of 14,000 Da.

#### 4.2.3. Physicochemical Characterization of NB Formulations

The mean hydrodynamic diameter, polydispersity index (PDI) and zeta potential of the NBs were measured by dynamic light scattering spectroscopy (DLS) at room temperature. The samples were diluted with ultrapure water in an electrophoretic cell. Each measured value was the average of ten, and an electric field of 15 V/m was used for zeta potential determination. Photon correlation spectroscopy (PCS) with a scattering angle of 90° and a temperature of 25 °C using a 90 Plus instrument (Brookhaven, NY, USA) was used.

The viscosity of the NB formulations was determined at 25 °C using an Ubbelohde capillary viscosimeter (Schott Gerate, Mainz, Germany). 

#### 4.2.4. TEM and US

The nanoscale morphological and chemical characterization of GC-DFO NBs before and after iron chelation was conducted through transmission electron microscopy (TEM) by using a JEOL 2200FS microscope working at 200 kV. NBs were sonicated by US clinical equipment (MyLab™25Gold Esaote, Genova, Italy) connected to a linear array transducer (LA523, 7.5 MHz central frequency, Esaote, Genova, Italy) operating in B-mode using the small parts imaging preset.

#### 4.2.5. Calculation of the DFO Grafted to the GC-DFO

The amount of DFO grafted was determined by titration with Fe^3+^ by measuring absorption at 430 nm (maximum wavelength of DFO-Fe^3+^ complex). The polymer was first dissolved in water at 0.2 gL^−1^. Then, various solutions of 0.1 gL^−1^ were prepared through the addition of increasing concentrations of Fe^3+^ chloride solution (80–12,800 µM). The absorption at 430 nm was plotted versus the iron content in solution, calculated in mmol per gram of polymer. The iron capacity of DFO was determined as the concentration at the observed change in the slope.

The slope discontinuity observed in Figure 8 at 8.83 mmol/g corresponds to the saturation of DFO sites with Fe^3+^. This DFO content per gram of polymer corresponds to a DFO grafting rate of w = 0.061. 

The mass fraction of the DFO-grafted repeat unit (*w*) was then calculated from the slope with the following equation:$$w=\frac{nDFO+Mz}{mTOT\cdot \left(1-wc\right)}$$
where nDFO is the mole number of the DFO-grafted repeat unit, mTOT is the (dry) mass of dissolved GC-DFO in solution, Mz is the molar mass of DFO-grafted repeat units and wc is the water content of the synthetized polymer.

The dimensionless absorbance was computed as the logarithm of the ratio of incident to transmitted radiant power. 

Statistical differences were evaluated using two-way ANOVA followed by Tukey’s multiple comparisons test.

#### 4.2.6. Thioflavin T Assay of Amyloid Formation

A fluorescence assay was performed using an automated multimode plate reader, EnSightTM, with a 440 nm excitation filter and 480 nm emission filter at 37 °C. The analysis was performed on 96-well plates with a total volume of 100 μM.

A reaction mixture was prepared with 20 μM thioflavin T, 5 μM Aβ_1–42_ and different FeCl_3_ concentrations in the range of 0.15–1.23 μM in modified Krebs–Henseleit buffer (123 mM NaCl, 4.8 mM KCl, 1.2 mM MgSO_4_, 1.4 mM CaCl_2_, 11.0 mM glucose and 100 mM PIPES, pH 7.4). The same reaction mixture was added to Fe^3+^ chelated with GC-DFO, DFO and NB-GCDFO. The plates were read after being shaken for 5 s. 

Statistical differences were evaluated using one-way ANOVA followed by Sidak’s multiple comparisons test.

#### 4.2.7. Hemolytic Activity

Hemolytic activity of GC-DFO NBs was evaluated using rat blood. The blood was diluted 1:10 with PBS (pH 7.4). Triton X-100 1% was used as a positive control, where red blood cell breakage and the further release of hemoglobin occurred. Saline solution (NaCl 0.9% *w*/*v*) was used as a negative control. A total of 1 mL of all samples was prepared (1:10, 1:100, 1:200 dilution) and incubated for 90 min at 37 °C with different concentrations of GC-DFO NBs (0.01 mg/mL; 0.02 mg/mL; 0.1 mg/mL). Then, samples were centrifuged for 5 min at 2000 RPM and the supernatant was analyzed with an ultraviolet–visible spectrophotometer (DU 730, Beckman Coulter, Fullerton, CA, USA) at 543 nm. The percentage of hemolysis was calculated using the positive control to represent 100% hemolysis.

#### 4.2.8. ROS Production Assessment

For ROS quantification, a fluorimetric method, based on the oxidation of the fluorescent probe 2′,7′-dichlorofluorescin, was used. For the 2′,7′-dichlorofluorescin test, a 400 µM H2DCF solution was prepared by mixing 0.5 mL of a 10 mM 2′,7′-Dichlorodihydrofluorescein diacetate (H2DCFDA) ethanolic stock solution with 2 mL of NaOH 0.01 M. The hydrolysis product, H2DCF, was kept at room temperature for 30 min and neutralized with 10 mL of 50 mM phosphate buffer (pH 7.4). In the presence of ROS, H2DCF was rapidly oxidized in the fluorescent DCF (2′,7′-dichlorofluorescin). The fluorescence intensity of all standards and samples was measured using a spectrofluorometer (EnSightTM automated multimode plate reader, Perkin Elmer, Waltham, MA, USA) at emission and excitation wavelengths of 485 and 530 nm. The concentration of ROS was calculated from a calibration curve of DCF. The calibration curve was obtained within a concentration range of 0.00122–0.62 µM and with an R^2^ of 0.9986. For ROS determination in the samples, a volume of H2DCF solution (400 µM) was added to each sample, previously diluted in water in a ration of 1:100, to obtain final concentrations of 5 µM. The ROS assays were performed by adding 50 µg/mL of Fe^3+^ to the first sample, and then, by adding GC-DFO NBs and Fe^3+^ prepared in a ratio of 1:1 (*w*/*w*) with Fe^3+^ (50 µg/mL) and GC-DFO (50 µg/mL) to the second sample. Samples were allowed to equilibrate at room temperature in the dark for at least 20 min, to allow the reaction to be completed. The intensity of fluorescence was measured before 60 min. All the experiments were performed in triplicate.

#### 4.2.9. Organotypic Brain Culture Preparation

Wistar Han TM Rats were obtained from the animal facility of the University of Trieste 5 days after birth (P5). Immediately after sacrifice by decapitation, the ventral tegmental area containing the substantia nigra (SN) was dissected and maintained in dissection medium (ice-cold Gey’s Balanced Salt Solution—Sigma-Aldrich, St. Louis, MO, USA, plus d-Glucose 10 mg/mL) until use. A McTwain tissue chopper (Gomshall Surrey, UK) was used to transversely cut 300 µm slices. Healthy slices were selected under stereomicroscope inspection. Slices were then transferred to sterile, semi-porous Millicell-CM inserts (PICM03050, Millipore, Darmstadt, Germany) in a 6-well plate, fed 1 mL of OBCs-SN medium (65% Basal Medium Eagle—Life Technologies Corporation, Grand Island, NY, USA; 10% heat-inactivated Fetal Bovine Serum—Euroclone, Milan, Italy; 25% Hank’s Balanced Salt Solution—Sigma-Aldrich, St. Louis, MO, USA; 1% L-Glutamine—Life Technologies Corporation, Grand Island, NY; 2% Penicillin/Streptomycin—Life Technologies Corporation, Grand Island, NY; 10 mg/mL D-Glucose—Sigma-Aldrich, St. Louis, MO, USA) and maintained at 37 °C, 5% CO_2_ and 95% humidity in a humidified incubator [37,38]. The medium was changed the day after cutting and every two days thereafter. 

#### 4.2.10. Treatments

Slices were maintained in culture 10 days before starting treatment, allowing them to recover from the stress of the slicing procedure [38]. Then, slices were treated for 24 h with rotenone (Rot, dissolved in DMSO, Sigma-Aldrich) and diluted to a final concentration of 10 nM in OBCs-SN medium. Rot was used to reproduce the dopaminergic neuron (DOPAn) loss present in PD (negative control) [38]. As an internal experimental control, slices were exposed to the same amount of DMSO required for dissolving Rot (positive control, reference) and NBs. Untreated slices were used to assess the safety of DMSO as a positive control. GC-DFO and GC-DFO (after chelation) were diluted in OBCs-SN medium. Dilutions (1:8 to 1:192) were chosen based on a previous work [39]. 

#### 4.2.11. Viability Tests

Three different tests were used to assess the potential effect of NBs on slices (mitochondrial activity, evaluation of membrane leakage and counting of dopaminergic neurons).

Mitochondrial activity

Mitochondrial metabolic activity was assessed, as previously described [37], using a 1-(4,5-dimethylthiazol-2-yl)-3, 5-diphenylformazan (MTT) assay (Sigma-Aldrich, St. Louis, MO, USA). At the end of experiments, slices for each repetition were incubated with 0.5 mg/mL of MTT in medium at 37 °C, 5% CO_2_ and 95% humidity in a humidified incubator for 1 h. Then, DMSO was added to dissolve the precipitated salts, and the absorbance of the solution was read at 562 nm in an EnSpire Multimode Plate Reader (PerkinElmer, Waltham, MA, USA). Results were expressed as fold vs. control (DMSO = 1).

2.Evaluation of membrane leakage

The amount of total extracellular LDH (lactate dehydrogenase) in the medium, indicative of membrane leakage, was determined using a CytoTox-ONE™ Homogeneous Membrane Integrity Assay (G7891, Promega, Madison, WI, USA), as previously described [37]. The signal was read at 560/590 nm (excitation/emission) by an EnSpire Multimode Plate Reader (PerkinElmer, Waltham, MA, USA). Results were expressed as fold vs. control (DMSO = 1).

3.Counting of Dopaminergic Neurons

DOPAns were counted using the immunofluorescence technique, as previously described [38]. At the end of challenging, slices were immediately fixed in 4% neutral buffered formalin for 30 min at 4 °C, then pre-incubated for 1 h in 10% normal donkey serum (Sigma-Aldrich), 1% bovine serum albumin (Sigma-Aldrich) and 0.1% Triton-X100 (Sigma-Aldrich) in PBS (blocking solution). To stain DOPAns, slices were incubated for three days at 4 °C with primary polyclonal antibody anti-tyrosine hydroxylase (TH+, 267 ng/mL, AB152, Millipore, Temecula, CA, USA) in an incubation solution containing 1% normal donkey serum (Sigma-Aldrich), 1% bovine serum albumin (Sigma-Aldrich) and 0.1% Triton-X100 (Sigma-Aldrich). After rinsing three times in PBS for 5 min, slices were incubated with donkey anti-rabbit Alexa fluor 546 secondary antibodies (1 µg/mL, A10040, Life Technologies, Carlsbad, CA, USA) overnight at 4 °C in incubation solution. Slices were then washed twice with PBS, stained with Hoechst 33258 (2 µg/mL, Sigma-Aldrich), washed with water and mounted (Dako, Agilent, Santa Clara, CA, USA). Hydroxylase-positive (TH+) DOPAns were counted using a Leica DM2000 fluorescence microscope (Leica Microsystems Srl, Solms, Germany) by three separate individuals. The results were expressed as percentages relative to the controls (DMSO, 100%).

#### 4.2.12. MEA: Micro-Electrode Arrays Recordings

Micro-electrode arrays (MEAs) were purchased from Multichannel Systems (MCS, Reutlingen, Germany). MEAs consisted of 60 TiN (titanium nitride) planar round electrodes (30 mm diameter; 200 mm center-to-center inter-electrode distance). An MEA amplifier was placed inside an incubator with a controlled temperature (37 °C) and humified atmosphere with 5% CO_2_. All measurements were performed by keeping the neurons in their culture medium. The acquired signals, after 1200× amplification, were sampled at 10 kHz and acquired through data acquisition hardware and MC-Rack 3.8 software (MCS). For each trial, data acquisition was performed throughout 2 min recordings.

Recordings started 5 min after drug/NB administration, to restore temperature and CO_2_ conditions inside the incubator [40].

Subsequent recordings were performed 5, 24 and 48 h after drug or NB administration.

Activity burst analysis [40] was performed using Neuroexplorer 4 software (Nex Technologies, Lenora, KS, USA) after spike sorting operations. A burst consists of a group of spikes with decreasing amplitude; thus, we set a threshold of at least 3 spikes and a minimum burst duration of 10 ms. We set interval algorithm specifications, such as maximum interval to start burst (0.17 s) and maximum interval to end burst (0.3 s), recorded in 0.02 s bins [40]. Burst analysis was performed by monitoring the following parameters: mean frequency and number of bursts. Data are expressed as means ± SEM, and statistical significance was calculated by using Student’s *t*-test. Values of *p* < 0.05 were considered significant. The number of experiments refers to the number of MEAs each of those consisting of 59 recorded electrodes.

#### 4.2.13. Hippocampal Neuronal Cultures

Hippocampal neurons were obtained from 18-day-old C57BL/6 mouse embryos. All animals were housed under a 12 h light/dark cycle in an environmentally controlled room with food and water provided ad libitum. The embryos were removed immediately by caesarean section and rapidly decapitated before the removal of tissue. The hippocampus was rapidly dissected under sterile conditions, kept in cold Hanks’ balanced salt solution (HBSS) (4 °C) with high glucose and digested with papain (0.5 mg/mL) dissolved in HBSS plus DNAse (0.1 mg/mL) as previously described [40]. Isolated cells were then plated at a final density of 100 cells/mm2 and maintained in a culture medium consisting of Neurobasal, B-27 (1:50 *v*/*v*), glutamine (1% *w*/*v*) and penicillin–streptomycin (1%) (all from Sigma-Aldrich, St Louis, MO, USA). Every 7 days for 2–3 weeks, a half-volume medium replacement was performed [40].

The normal distribution of experimental data was assessed with the D’Agostino–Pearson normality test. Unless otherwise specified, the statistical significance of our data was evaluated by considering two sample groups that were normally distributed and applying Student’s *t*-test. Otherwise, in cases of more than two sample groups that were normally distributed, one-way ANOVA followed by Bonferroni’s test was used. In cases of two sample groups that were not normally distributed, we used the non-parametric Mann–Whitney’s U-test or, when indicated, the Kolmogorov–Smirnov test. The Kruskal–Wallis test followed by Dunn’s post hoc test was used in cases of more than two sample groups that were not normally distributed. Data were considered statistically significant when *p* < 0.05.

## 5. Conclusions

The results of this study could be relevant since cognitive impairment in neurodegeneration, such as AD and PD, is strictly related to the loss of neuronal connection, partly due to the mechanical constraint of the plaques, but also, the impairment of the bioelectrical signaling can be reversed by effective local iron chelation. 

In conclusion, the results of our extensive in vitro investigations of GC-DFO NBs are very encouraging, and their application to in vivo models will be a challenging future task.

## 6. Patents

A nanostructure for the vehiculation of gas and/or active ingredients and/or contrast agents and uses thereof, PCT extension, WO2015/028901 A1.

## Figures and Tables

**Figure 1 ijms-25-00729-f001:**
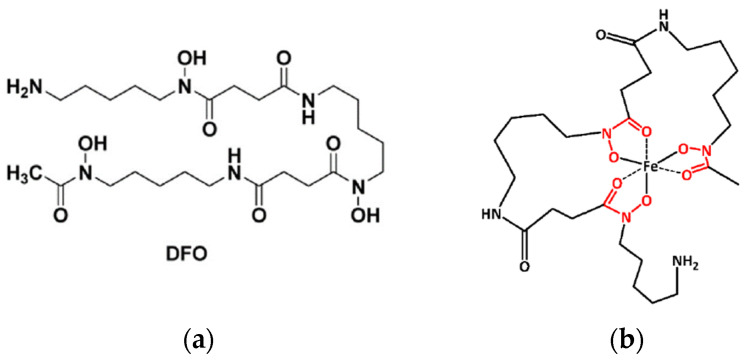
(**a**) Linear structure of DFO (**b**) and its iron chelation capacity at all 6 coordination sites.

**Figure 2 ijms-25-00729-f002:**
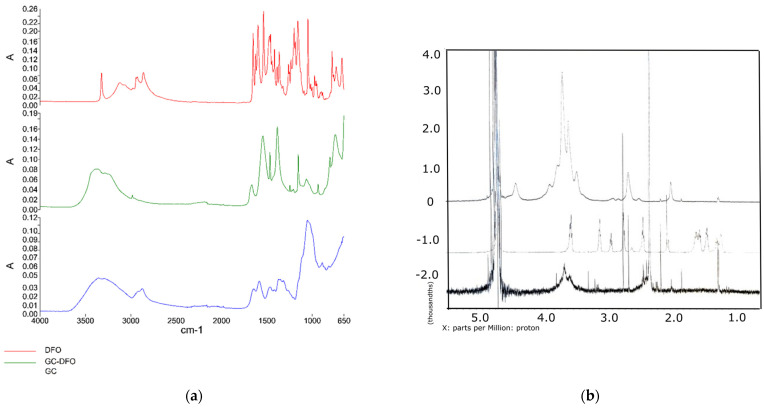
(**a**) FT-IR of GC (in blue), GC-DFO conjugate (in green) and DFO (in red); (**b**) NMR of GC (**upper**), DFO (**intermediate**) and GC-DFO conjugate (**lower**).

**Figure 3 ijms-25-00729-f003:**
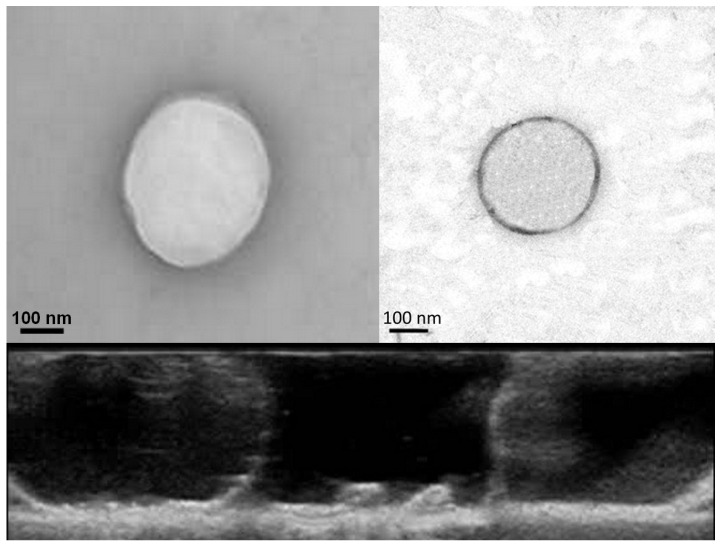
In the upper part, TEM images for GC-NB (**left**) and GC-DFO NB (**right**). In the lower part, image from US for GC-NB (**left**) and GC-DFO NB (**right**).

**Figure 4 ijms-25-00729-f004:**
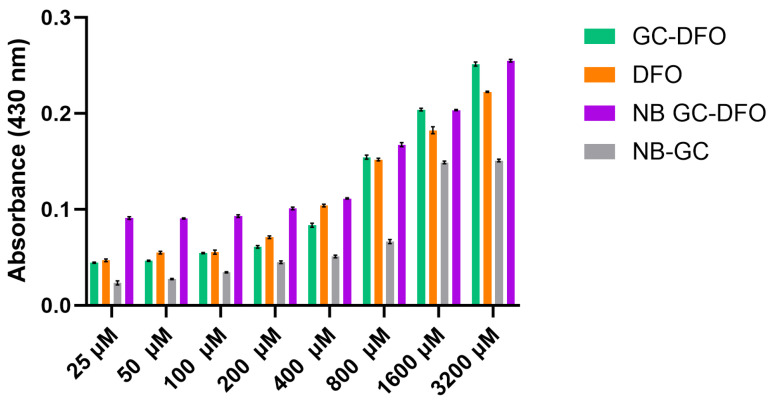
Comparison between chelation ability of DFO (orange), GC-DFO conjugate (green), NB GC (grey) and NB GC-DFO (purple) at increasing iron concentrations (µM) measuring absorption at 430 nm.

**Figure 5 ijms-25-00729-f005:**
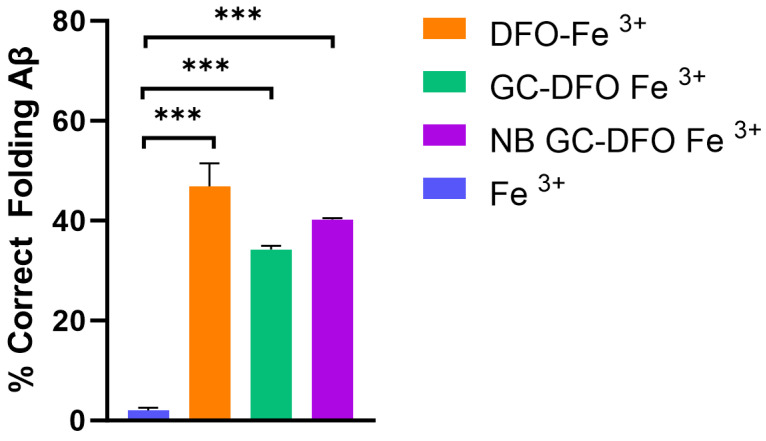
Percentage of corrected folded Aβ in the presence of Fe^3+^ and different iron chelators computed from the fluorescence intensity (RFU). *** *p* < 0.001, one-way ANOVA followed by Sidak’s multiple comparisons test.

**Figure 6 ijms-25-00729-f006:**
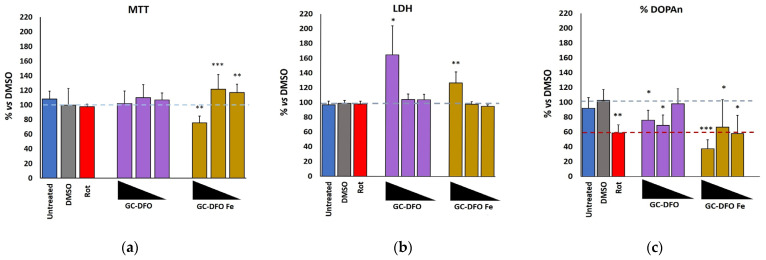
Percentage viability of cells after 24 h of treatment. (**a**) MTT, (**b**) LDH and (**c**) DOPAn counts of the selected nanoformulations (GC-DFO = pre-chelation, GC-DFO Fe = post-chelation). Data are expressed as % vs. control (DMSO). Concentrations of DFO corresponding to the 3 bars above the black triangles are from left to right: dilution of 1:8 = 125 μg/mL, 1:64 = 15.62 μg/mL, 1:192 = 5.21 μg/mL. N = 6 biological repetitions. Untreated (blue); DMSO (gray); rotenone (red); GC-DFO NBs (purple); GC-DFO NBs post-chelation (dark yellow). For reference, gray dashed lines are DMSO and red dashed line is rotenone (Rot). * *p* < 0.05; ** *p* < 0.01; *** *p* < 0.001.

**Figure 7 ijms-25-00729-f007:**
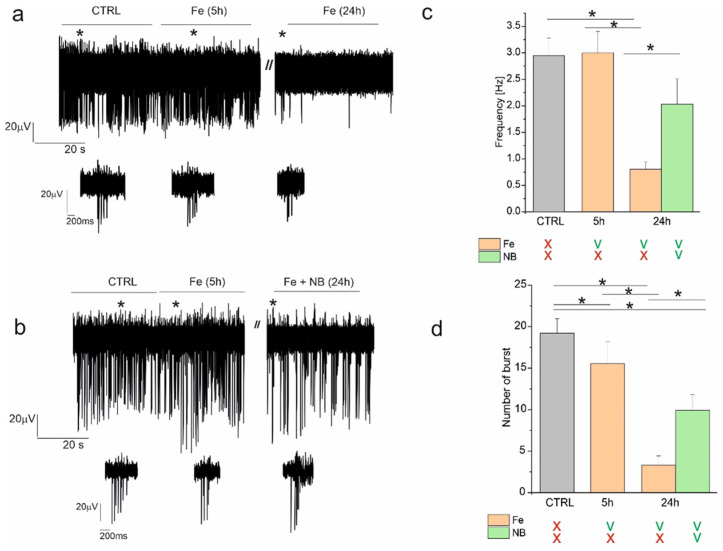
NBs revert the impairments caused by Fe administration in the spontaneous electrical activity of the hippocampal neuronal network, measured using MEAs. (**a**) One representative trace of spontaneous firing recorded with MEAs in the control condition (CTRL), after 5 and 24 h of incubation with Fe, compared (**b**) to that observed when spontaneous firing was recorded after 24 h of incubation with Fe and NBs. The asterisks indicate one representative burst at the expanded scale shown in the insets below the traces. (**c**) Bar graph of the average firing frequency recorded in control conditions (CTRL) (grey), after 5 h of incubation with Fe (orange) and after 24 h of incubation with Fe, alone (orange) or together with NBs (green). (**d**) Bar graph of the average number of bursts recorded in the same experimental conditions shown in (**c**). Below the bar graphs, the absence (X) or presence (V) of Fe and/or NBs during the different experimental conditions is indicated. * *p* < 0.05.

**Figure 8 ijms-25-00729-f008:**
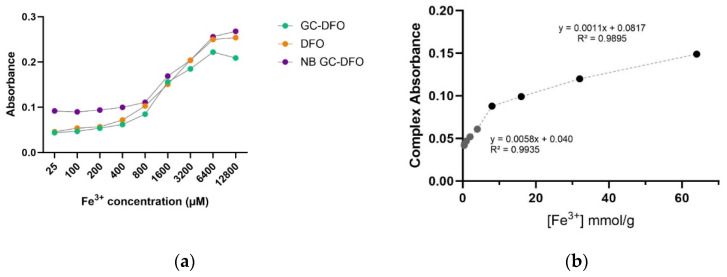
(**a**) Comparison of iron chelation for different nanoformulations at different concentrations; (**b**) iron capacity of DFO, determined based on observed changes in the slope.

**Table 1 ijms-25-00729-t001:** Physicochemical characterization of different formulations of NBs.

NBs Formulation	Average Diameter (nm)	Polydispersity Index (PDI)	Zeta Potential (mV)
Glycol-chitosan	228.7 ± 3.8	0.18 ± 0.06	+20.15 ± 1.69
GC-DFO	166.6 ± 23.2	0.18 ± 0.08	−39.48 ± 4.71
GC-DFO-Fe^3+^	143.2 ± 2.27	0.28 ± 0.03	−23.21 ± 2.29

## Data Availability

The data presented in this study are available on request from the corresponding author.

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
