# Peer review of "Developing Iron Nanochelating Agents: Preliminary Investigation of Effectiveness and Safety for Central Nervous System Applications"

_ijms, 2024, doi:10.3390/ijms25020729_

Round 1

Reviewer 1 Report

Comments and Suggestions for Authors

I reviewed the ms "Iron Nano-Chelating Agents: Preliminary Investigation on Effectiveness and Safety for Central Nervous System applications" by Ficiarà et al. about iron-chelating nanobubbles.

Unfortunately I must reject this submission, as the article has several problems (controls missing in several experiments, poor explanation of both background and results, poor discussion, crucial data not shown, incongruencies between figures and text). 

I suggest authors thoroughly rearrange their work.

Comments on the Quality of English Language

Some parts are difficult to follow, and grammar is also an issue in some sentences.

Author Response

We thank the reviewer for his timely and relevant comments, we responded to all the observations in the rebuttals trying to improve all aspects: the introductory and discussion parts, the figures; we better specified the methods, and added all the relevant control conditions. We are sure that after these changes, which also required additional experimental work, the paper has improved significantly.

Point-by-point answers are in the attached pdf.

Reviewer 2 Report

Comments and Suggestions for Authors

The authors proposed an advanced nanobubble formulation with iron chelating capacity, investigating the main physicochemical properties and the relative safety for potential applications in neurodegenerative diseases. The manuscript is well written and structured, the introduction provides sufficient background. The authors obtained interesting results. However, some questions require clarification.

1. Line 65 – 67: lacks a corresponding literature reference.

2. Line 85: “the central nervous system (CNS)…” the abbreviation has already been introduced above.

3. It seems necessary to formulate the aim of the study in the Introduction.

4. Line 85: “slices Organotypic Brain Cultures-Substantia Nigra…” N values should be added.

5. Fig. 5 and Fig. 6: Please make more detailed captions.

6. Line 300 – 305 and below: it seems necessary to add links to the figures, as done above.

7. 4.2.8 Organotypic Brain Culture Preparation and 4.2.10 Viability tests: please add the appropriate references.

8. Line 316 – 317: Have similar studies been conducted on surviving slices before? It seems necessary to include the answer in the Discussion.

9. Line 501: “slices were incubated three days at 4oC with primary polyclonal antibody…” please explain why it took so long.

Author Response

Thank you for your thoughtful and constructive feedback on our manuscript. We appreciate your positive comments on the clarity and structuring of the writing, and the interesting results obtained. We also acknowledge the importance of addressing the questions that require clarification. We are committed to providing thorough responses to your queries, and we will revise the manuscript accordingly to enhance its overall quality.

Point-by-point response is in the attached pdf.

Round 2

Reviewer 1 Report

Comments and Suggestions for Authors

I can see the amount of work authors put to ameliorate their manuscript, and I think this is now much better. A lot of work was not presented but now it's explained and described, so I'm okay with publication. 

Reviewer 2 Report

Comments and Suggestions for Authors

I'm okay with publication